# Permian hypercarnivore suggests dental complexity among early amniotes

Tea Maho[1,2], Sigi Maho[2], Diane Scott[2] & Robert R. Reisz [1,2] ✉

The oldest known complex terrestrial vertebrate community included hyper-carnivorous varanopids, a successful clade of amniotes with wide geographic and temporal distributions. Little is known about their dentition and feeding behaviour, but with the unprecedented number of specimens of the varanopid *Mesenosaurus* from cave deposits in Oklahoma, we show that it exhibited serrations on the tooth crowns, and exceptionally rapid rates of development and reduced longevity relative to other terrestrial amniotes. In contrast, the coeval large apex predator *Dimetrodon* greatly increased dental longevity by increasing thickness and massiveness, whereas herbivores greatly reduced tooth replacement rates and increased dental longevity. Insectivores and omnivores represented the primitive condition and maintained modest replacement rates and longevity. The varied patterns of dental development among these early terrestrial amniotes reveal a hidden aspect of dental complexity in the emerging diverse amniote community, very soon after their initial appearance in the fossil record.

Most vertebrates replace their teeth continuously throughout ontogeny, a condition known as polyphyodonty[1,2]. This is commonly accomplished by sustaining a stem cell population in the dental lamina, lingual to the functional tooth, that can develop into replacement teeth[3–5]. Recent evidence suggests that this may represent the primitive condition of all jawed vertebrates[6]. Evolutionary changes and variation in tooth replacement patterns are intimately associated with patterns of higher vertebrate diversification and evolution, changes occurring at various stages of amniote evolution, where tooth replacement has been either extensively modified, reduced, or entirely eliminated[7].

Early amniotes are also characterized by polyphyodonty. A greater understanding of the rates of tooth development and replacement in these ancient Paleozoic relatives of extant crown amniotes is likely to provide valuable insights into some of the mechanisms associated with the evolution of the complex feeding behaviours and ecologies that characterize the diversification of reptiles, the dentigerous relatives of birds, and mammals. Dental characteristics, including patterns of replacement, may be used to infer aspects of an organism's lifestyle[8–10], including adaptations to herbivorous, omnivorous, or even highly carnivorous diets[11]. This is particularly

important because early amniotes rapidly adapted to a wide range of feeding strategies, including insectivory, herbivory with and without dental occlusion, piscivory, omnivory, and various levels of carnivory[12].

Permo-Carboniferous amniote fossils record the deep phylogenetic divergence between synapsids and reptiles, the former represented by pelycosaur-grade synapsids[13], the latter represented by eureptiles and parareptiles. Among early synapsids, varanopids occupy a unique position in having the longest fossil record (Supplementary Information), extending from the late Carboniferous into the late middle Permian (more than 40 million years), a near-global distribution[14–16], and the earliest evidence of parental care[17]. Permo-Carboniferous amniotes are generally rare and not readily available for destructive histological studies. However, recently discovered abundant material of the varanopid *Mesenosaurus efremovi*[14] (Fig. 1) at the Richards Spur Lagerstätte (a deposit that exhibits extraordinary fossil richness and exceptional preservation) allowed us to undertake a detailed investigation of the dental development and replacement in this early Permian varanopid, and place it within the broader context of feeding strategies in the oldest known complex terrestrial vertebrate community.

[1]International Centre of Future Science, Dinosaur Evolution Research Center, Jilin University, Changchun, Jilin 130012, China. [2]Department of Biology, University of Toronto Mississauga, Mississauga, ON, Canada. ✉e-mail: robert.reisz@utoronto.ca

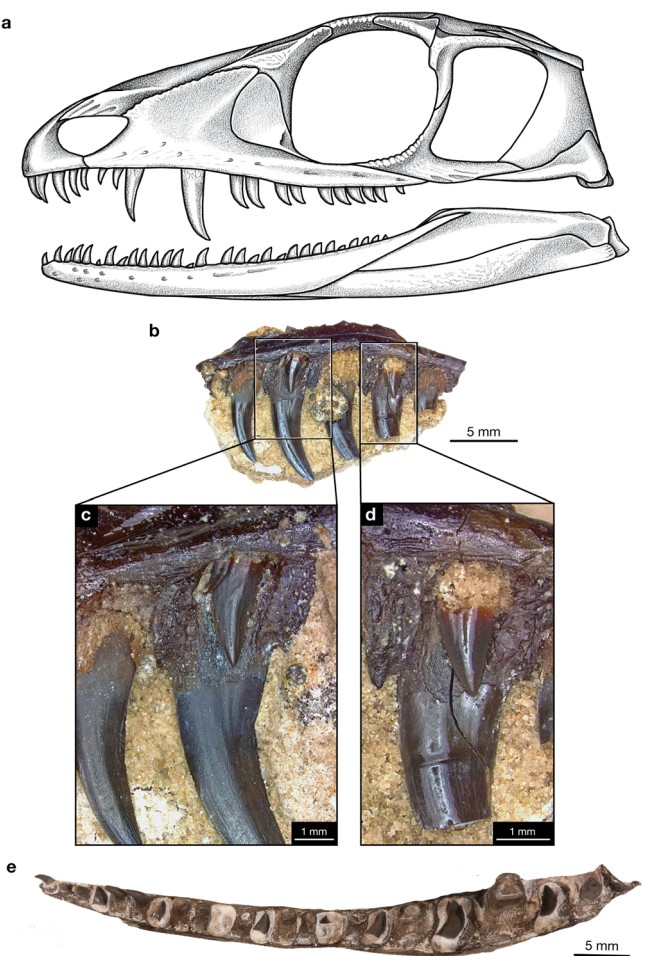

**Fig. 1 | Tooth replacement in maxillary dentition of *Mesenosaurus efremovi*.**
**a** Reconstruction of *M. efremovi* skull in lateral view. Note that gaps in dentition represent shed teeth that are being replaced. **b** Maxillary dentition of ROMVP 85456 in lingual view. **c** Tooth family with a functional tooth and two successive replacement teeth. **d** Tooth family with a functional tooth and a single replacement tooth. **e** Photograph of maxilla (ROMVP 85524) in occlusal view showing the pattern of tooth replacement, with new teeth developing in the sockets of old teeth after the latter were shed.

Tooth replacement rate is defined here as the difference between the total number of days recorded in a functional tooth (i.e., the tooth's age from the start of the mineralization of its dentine) and the total number of days in the successive replacement tooth for a tooth position[18], or the age of the functional tooth when a tooth replacement event is initiated; thus, a new tooth starts to deposit dentin. Tooth development and replacement in both extinct and extant polyphyodont organisms can be assessed by studying incremental growth lines of von Ebner within dentine[18,19], the main component of teeth. In extant amniotes, lines of von Ebner are dentinal growth lines formed through the daily mineralization of dentine in apparent response to circadian rhythms; each line reflects the daily formation of dentine by odontoblasts which occurs continuously, unless the organism experiences metabolic disruptions[18–27]. Using periodic chemical labelling, Erickson[18,19] demonstrated that lines of von Ebner formed daily in extant crocodilians and that the teeth of fossil crocodilians possessed morphologically equivalent lines to those found in dinosaur teeth. This homologous growth pattern among all amniotes[18,19,28] provides the means to study tooth development and replacement in extinct vertebrates, including early synapsids and reptiles.

This study examines the evolution of dental development and replacement rates in early amniotes and its relationship with the species' dietary specialization for members of the oldest known complex terrestrial vertebrate community. Additionally, the study identifies the incremental lines observed within the dentition of the taxa as being the incremental lines of von Ebner and calculates the replacement rates of the dentition. With the availability of a rich data set, we also assess comparatively the overall tooth longevity, an important aspect of dental life history and feeding strategy.

*Institutional abbreviations*—**OMNH**, Sam Noble Oklahoma Museum of Natural History, University of Oklahoma, Norman, OK, USA; **PIN**, Paleontological Institute, Russian Academy of Sciences, Moscow, Russia; **UCMP**, University of California Museum of Paleontology, Berkeley, USA; **USNM**, National Museum of Natural History, Smithsonian Institution, Washington, DC; **ROMVP**, Royal Ontario Museum, Toronto, ON, Canada.

## Results

All vertebrates examined in this study and histologically sampled (Supplementary Table 1) exhibit polyphyodonty and dentine growth lines (Figs. 2–4 and Supplementary Figs. 2–9) that are morphologically consistent with the incremental lines of von Ebner of extant mammalian and crocodilian teeth: alternating opaque zones, line trajectories paralleling the pulp cavity, and widths ranging between 1 and 30 mm[18]. All functional teeth were continuously replaced through the development of the replacement tooth, lingual to the functional tooth, resulting in resorption of its base and shedding.

### Replacement pattern in *Mesenosaurus efremovi*

Replacement in the gracile predator *Mesenosaurus efremovi* from the Richards Spur locality (Fig. 1) appears to occur as a wave in alternating tooth positions, with every other functional tooth in a sequence undergoing replacement during one event. Gaps in the tooth row represent stages in the replacement cycle when the old tooth has been shed, but the replacement tooth has not yet become functional and is not ankylosed to the jawbone. Frequently, these small replacement teeth are lost during fossilization, but in the case of the Dolese *Mesenosaurus*, preservation is so exquisite that these unattached replacement teeth are preserved, often in place (Fig. 1e). We found that numerous specimens of *M. efremovi* have tooth families containing a functional tooth and a single replacement tooth lingual to it, but one maxilla (ROMVP 85456) was observed to have a tooth family containing a functional tooth and two successive replacement teeth (Fig. 1c).

The replacement rate found in one tooth family within an *M. efremovi* dentary was 39 days (ROMVP 85502; Fig. 2), and 34 days for the left maxilla (ROMVP 85443; Supplementary Fig. 2). Replacement rates of three tooth families (mx10, mx12, and mx15) for ROMVP 85457 were estimated to be 46, 36, and 35 days. Thus, the replacement rate for *M. efremovi* does not appear to vary significantly in one specimen across tooth position, size, or ontogenetic age of tooth.

### Replacement pattern in other synapsids

In contrast to the availability of many *Mesenosaurus* specimens for destructive sampling, other taxa are exceedingly rare, and few specimens were available for destructive analysis. Thus, only a single maxilla of the apex predator *Dimetrodon* with a replacement tooth in position was available (Fig. 3). The functional tooth had a total of 459 incremental lines, whereas the replacement tooth had a total of 354 lines, resulting in a replacement rate of 105 days. In contrast, the maxillary tooth for the basal sphenacodont *Haptodus*, was calculated to have functional tooth longevity of approximately 152 days and since neither a replacement tooth nor a resorption pit was present, the minimum replacement rate is 152 days.

Similarly, relatively little material was available for the larger varanopid predator *Watongia meieri* which is only known from the holotype material, with a resorption pit on one of the two teeth (mx19) on a

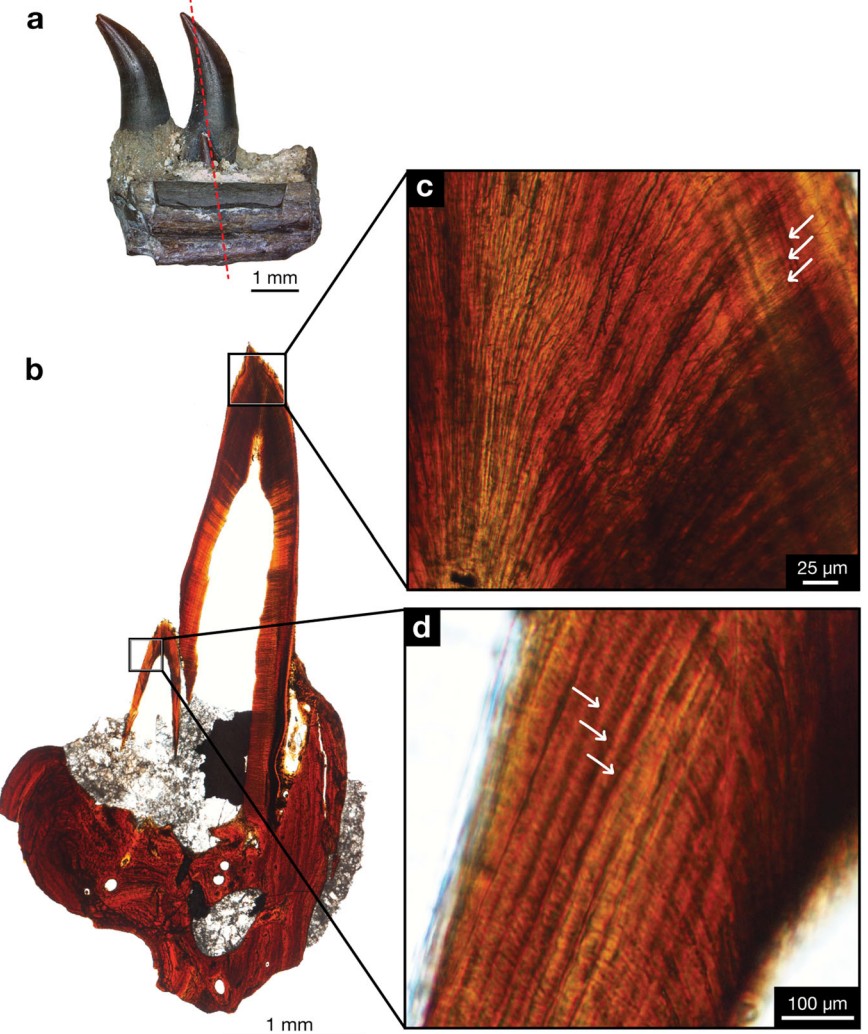

**Fig. 2 | Incremental lines of *Mesenosaurus efremovi*. a** ROMVP 85502, lingual view of fragmented dentary with dashed red lines through the plane of the LL section of the functional and replacement teeth. **b** Whole view of tooth family LL section near maxillary fragment, but both teeth were missing the crown apex; thus, crown apex. **c** Closeup view of functional tooth LL cross-section showing incremental lines, white arrows. **d** Closeup view of replacement tooth TR cross-section showing incremental lines, white arrows.

only a minimum age could be determined using the incremental line counts. The tooth with the resorption pit was determined to be a minimum of 81 days old, while the adjacent tooth not in the process of being replaced was approximately 68 days old. A second maxillary tooth with a resorption pit at mx18 was determined to be 145 days old. Additionally, one complete tooth with no resorption pit was longitudinally LL sectioned and estimated to be 108 days old.

One maxilla of the small, very rare herbivorous caseid *Oromycter* was available for destructive sampling (Supplementary Fig. 3). The tooth with a resorption pit in position mx07 was determined to have a total of 506 incremental lines, whereas the tooth without a resorption pit (mx09) had a total of 426 incremental lines. For the mx09 tooth family, the missing replacement tooth was estimated to have 115 incremental lines, resulting in an approximate replacement rate of 391 days.

The left dentary of the large herbivorous caseid *Ennatosaurus*, known only from five specimens, exhibited two posterior teeth with resorption pits on positions d08 and d07 (Supplementary Fig. 4). Tooth position d08 had a visibly larger and more developed resorption pit, with the functional tooth having a total of 628 incremental lines, whereas d07 had a smaller resorption pit and a total of 567 incremental lines. The missing replacement teeth for both d07 and d08 were

estimated to have 136 and 169 incremental lines, resulting in a replacement rate of approximately 431 and 459 days, respectively.

One maxilla of the herbivorous edaphosaurid *Edaphosaurus* had a resorption pit at tooth position mx09 (Fig. 4) and was estimated to have a total of 506 incremental lines. The adjacent tooth at position mx10 had no resorption pit and was determined to have a total of 429 lines. For the mx09 tooth family, the missing replacement tooth was estimated to have 131 incremental lines, resulting in a replacement rate of 381 days.

## Replacement pattern in early and extant reptiles

For the insectivorous parareptile *Delorhynchus* the functional tooth had a total of 147 incremental lines, while the replacement tooth had 43 lines (Supplementary Fig. 5), resulting in a replacement rate of 104 days. For the other parareptile *Colobomycter* the premaxillary functional tooth had a total of 157 incremental lines, whereas the replacement tooth had a total of 59 lines, resulting in a replacement rate of 98 days (Supplementary Fig. 6). For the omnivorous eureptile *Captorhinus*, the functional tooth was 146 days, and the replacement tooth was 69 days, resulting in a replacement rate of approximately 77 days. For the other eureptile, the highly specialized insectivore *Opisthodontosaurus*, the maximum tooth age for positions d04 to d07 was 151, 155, 206, and 258, respectively (Supplementary Fig. 7).

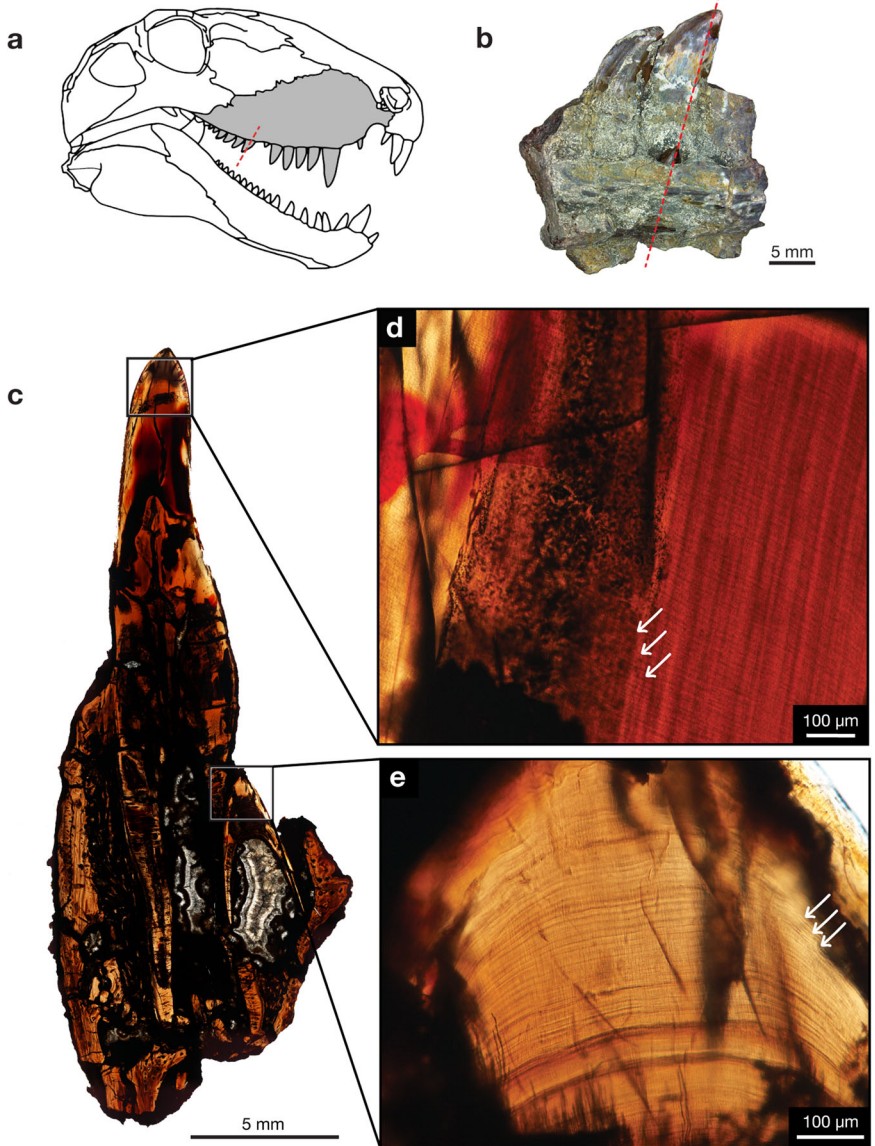

**Fig. 3 | Incremental lines of *Dimetrodon* cf. *D. limbatus*. a** Lateral view of *Dimetrodon*. **b** ROMVP 85510, maxillary tooth family, photographed in lingual view showing the plane of LL section through the functional tooth and replacement tooth. **c** Whole view of longitudinal LL section near the crown apex of functional and replacement tooth. **d** Closeup view of functional tooth LL cross-section showing incremental lines, white arrows. **e** Closeup view of replacement tooth LL cross-section showing incremental lines, white arrows. Skull drawing was modified from Reisz[42] and Brink and Reisz[43].

Although no replacement teeth were present, it was possible to use the resorption pit heights to estimate the replacement rates of 182 and 193 days for d06 and d07, respectively. These rates, although different from *Captorhinus* are not unexpected since this small, close relative of *Captorhinus* has a very odd, unusual dentition, specialized for feeding on harder shelled invertebrates.

In addition to the above Paleozoic amniotes, two skulls were examined for the extant varanid lizards, *Varanus bengalensis* and *Varanus komodoensis*, as well as shed teeth of the latter were also available for study and comparison. The maxillary bone of *Varanus bengalensis* carried dentition showing six replacement events, but only the mx04 tooth position was sectioned. The functional tooth was determined to have 188 incremental lines, and since a continuous record for the replacement tooth's incremental lines was not visible, the replacement rate was estimated based on its entire dentine area divided by the functional tooth's mean line width. The estimated replacement rate for *V. bengalensis* was approximately 110 days. Unlike *M. efremovi*, the base of the teeth is characterized by plicidentine, and

neither tooth serrations (ziphodonty; Supplementary Fig. 8) nor resorption pits were observed for *V. bengalensis*.

Similar to *Mesenosaurus*, *Varanus komodoensis*, a highly endangered varanid lizard, exhibits ziphodonty on both the mesial and distal tooth surfaces and provides a valuable comparison with the fossil taxon. Two isolated teeth of an adult individual that were in the process of attachment, but not yet ankylosed with the jaw-bone, were sectioned. The age of the first tooth was determined to have 106 lines, and the second tooth had approximately 135 lines. A third isolated shed tooth (due to resorption from replacement tooth or from the processing of food)[29] provided by the Toronto Zoo was determined to have approximately 227 incremental lines. Thus, from the age of initial tooth attachment to the age of shedding, a tooth appears to be functional for an average of 107 days. Additionally, as in *Mesenosaurus*, the adult skull of *V. komodoensis* (ROM R7565) showed that each tooth position exhibited multiple replacement teeth for both the dentary and the maxilla, also confirmed by the data from Auffenberg[30].

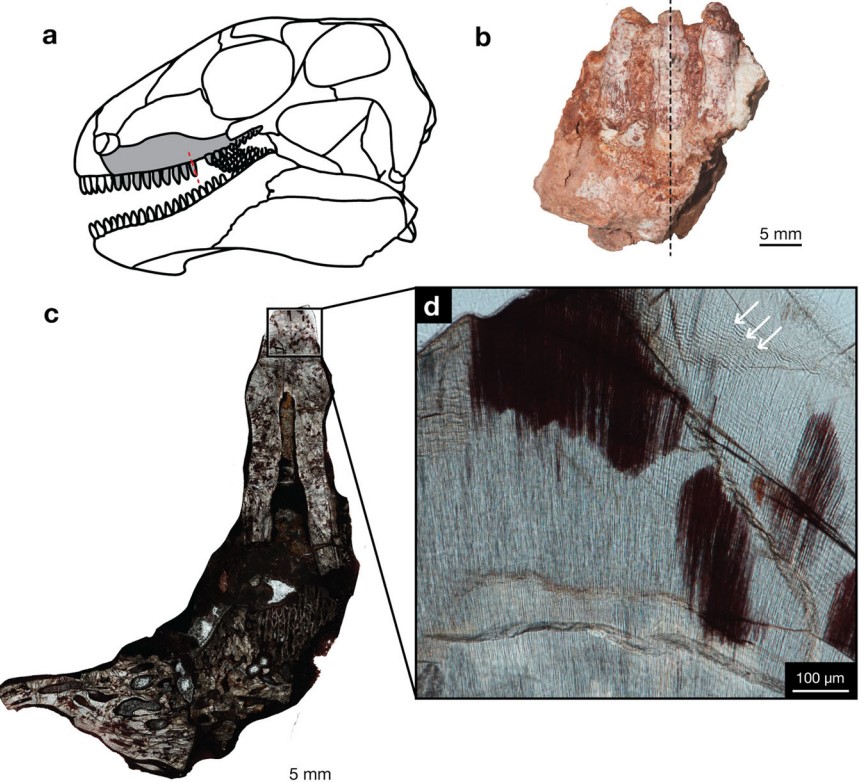

**Fig. 4 | Incremental lines of *Edaphosaurus* sp. a** Lateral view of *Edaphosaurus*. **b** USNM PAL 706602, maxillary tooth family, photographed in lingual view showing the plane of LL section through the functional tooth and replacement tooth. **c** Whole view of longitudinal LL section near crown apex of functional and replacement tooth. **d** Closeup view of functional tooth LL cross-section showing incremental lines, white arrows. Skull drawing was modified from Romer and Price[41] and Modesto[44].

### Replacement pattern in a stem amniote

For the representative carnivorous stem amniote *Seymouria* (Supplementary Fig. 9) the functional tooth was determined to have a maximum of 171 incremental lines, while the missing replacement tooth was estimated to have had approximately 36 lines. Thus, the estimated replacement rate for *Seymouria* was calculated to be 135 days.

### Replacement rate and body mass

There seems to be no significant relationship between replacement rate and body mass (kg) for the taxa examined (Supplementary Fig. 10). Although the largest body sized taxon *Ennatosaurus* had the longest replacement rate, but the other large species had varying rates, while the smallest taxa (*Captorhinus*, *Delorhynchus*, *Colobomycter*, and *Opisthodontosaurus*) all have varying replacement rates. Instead, replacement rates appear to be related to feeding behaviour since the herbivorous synapsids all exhibited long replacement rates and great tooth longevities (Fig. 5).

## Discussion

Previous studies have shown that incremental lines of von Ebner result from physiological processes of daily dentine deposition and mineralization[18,19,28]. The morphological similarity between incremental lines observed in Paleozoic stem amniotes, early reptiles, and synapsids presented here, with those of Mesozoic dinosaurs, crocodilians, and mammals, provides overwhelming evidence that this is a common growth pattern within Amniota[18,19,28,31].

Our results represent the first documentation of incremental lines of von Ebner, estimation of replacement rates, and tooth longevity for the marginal dentition of Paleozoic terrestrial amniotes. The taxa included here are some of the oldest known insectivorous, omnivorous, and herbivorous vertebrates, the oldest

terrestrial apex predator, as well as the medium-sized hypercarnivorous predator *Mesenosaurus*. It is important to note that previous work has suggested that reptilian replacement rates appear to be somewhat ontogeny dependent, with very early juveniles showing a higher replacement rate than later stages in ontogeny[19]. However, we restricted the selection of specimens to mostly subadults and adults for this study. This study provides additional taxonomic and dental data to the current phylomorphospace developed by Finch and D'Emic which examined the daily dentine apposition rate (DDAR) and body mass of species[32]. There seems to be no significant relationship between replacement rate and body mass (kg) for the early terrestrial vertebrates examined (Supplementary Fig. 10), similar to DDAR[32]. Feeding behaviour appears to have the greatest impact on replacement rate. However, to gain a better understanding of the variation in rates multiple factors should be considered in addition to feeding behaviour, such as prey preferences, morphology of the dentition, and body mass.

The early Permian varanopid *Mesenosaurus efremovi* stands out immediately for its exceptionally rapid tooth replacement and for the reduced longevity of the functional teeth, averaging approximately 41 days and 75 days, respectively. The strongly recurved, labiolingually compressed, and serrated crowns of this medium-sized predator (skull length of 10 cm) can be interpreted to reflect hypercarnivory with a diet comprising most likely of other tetrapods. Hypercarnivory (defined as consisting of at least 70% meat) has typically been used to describe the meat-only diets of extant mammals, like members of Carnivora. The coeval terrestrial apex predator, *Dimetrodon*, can also be considered to be a hypercarnivore, but there are some startling differences between these two predators in their tooth development and replacement patterns, revealing a hidden diversity in feeding behaviour.

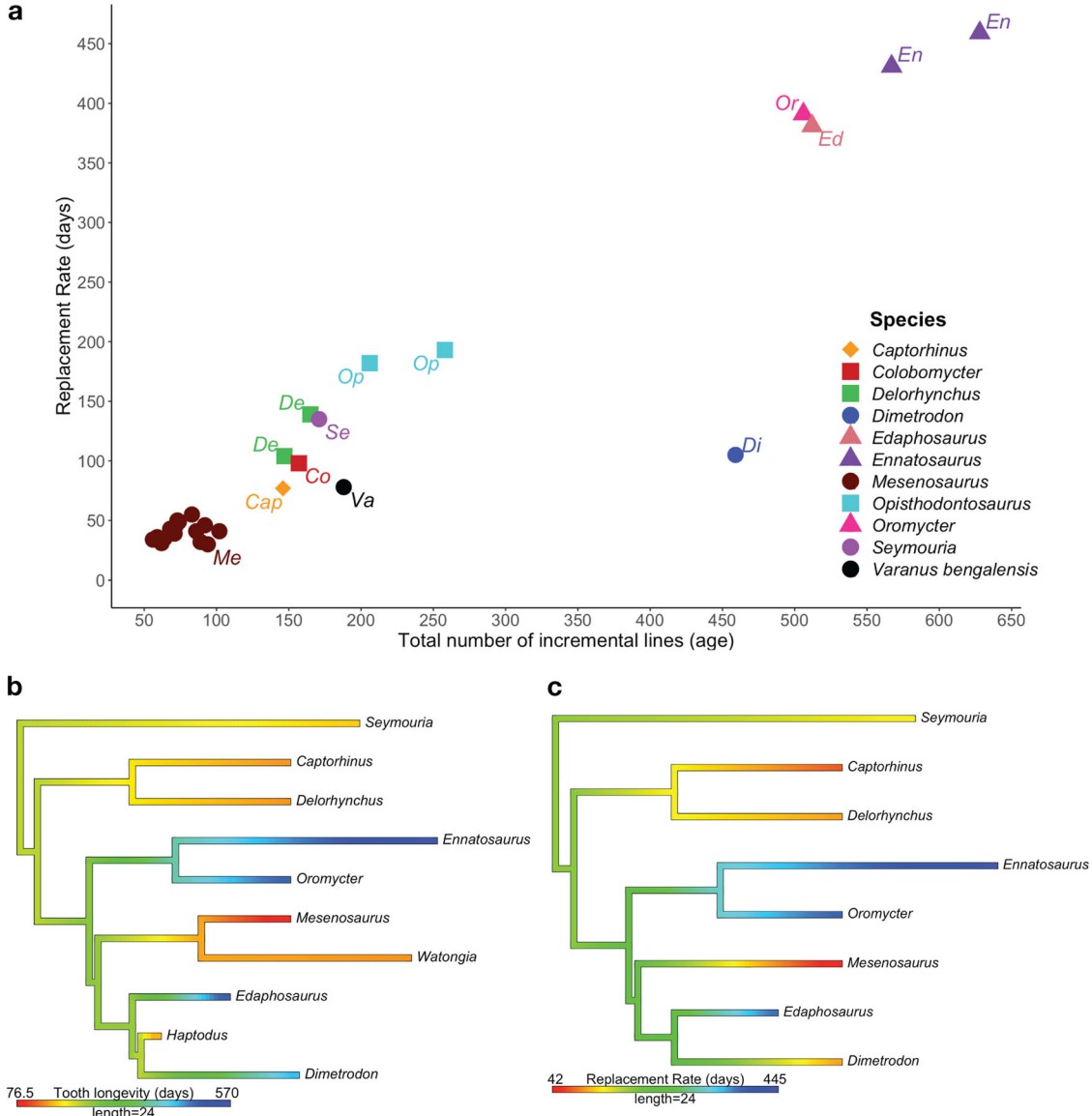

**Fig. 5 | Rates of tooth replacement and age across a range of taxa. a** Relationship between the total number of incremental lines of von Ebner (age) for the functional tooth and the tooth families replacement rate or period (days). The symbols indicate the type of feeding behaviour, with circles representing carnivory, triangles representing herbivory, square representing insectivory, and diamond representing omnivory. **b** Phylogenetic tree of all taxa (*n* = 11) used in the analyses, displaying the age in millions of years ago (length of bars) and tooth longevity (gradient in branch colours). **c** Phylogenetic tree of all taxa (*n* = 9) used in the analyses, displaying the age in millions of years ago (mya) (length of bars) and tooth replacement rate (gradient in branch colours). Reconstructed using the 'contMap' function in the 'phytools' R package. The tree was modified from Maddin, Evans, and Reisz[45] and Reisz and Sues[12]. Source data are provided as a Source Data file.

The estimated replacement rate (time to initiation of development for the new tooth) is approximately 105 days for *Dimetrodon*, but the replacement tooth (and not yet functional) was approximately 354 days old, showing that *Dimetrodon* not only produced new teeth less frequently than *Mesenosaurus*, but they also developed more extensively prior to the shedding of the functional tooth (longevity of 459 days). This apparent evolutionary innovation allowed for the formation and maintenance of thicker dentine and more massive teeth in *Dimetrodon* that would have served well for an apex predator. The functional tooth longevity for *Dimetrodon* appears to be an autapomorphy for this derived sphenacodontid since the basal sphenacodont *Haptodus* was found to have tooth longevity more in line with early Permian reptiles, approximately 152 days. In contrast to *Mesenosaurus efremovi*, the large varanopid *Watongia meieri* appears to have thicker incremental line widths (10–16 μm), showing an increase in daily dentine

deposition, but lacks the longevity of the teeth in *Dimetrodon*. *Watongia* is similar in body size to *Dimetrodon*, which could have resulted in it being the apex predator for its specific terrestrial vertebrate community[33].

The herbivorous synapsids appear to have even longer-lived teeth (430–630 days) and slower replacement rates (380–450 days), most likely due to their feeding behaviours. This is an interesting discovery because these herbivores show similar strategies for tooth replacement rates and dental longevity despite having slightly different food processing strategies. The caseids have teeth somewhat reminiscent of some squamates with the presence of leafy serrations on the crown and no evidence of tooth-on-tooth dental wear, and therefore limited oral processing. In contrast, edaphosaurs have tooth-on-tooth contact and wear on both the marginal dentition and the massive dental batteries on their palate and lower jaw, strong indicators of extensive oral processing[12].

In dinosaurs, the only other group of fossil vertebrates where tooth replacement patterns have been studied extensively, rates have also been found to be variable and somewhat related to diet and feeding behaviour. However, in contrast to Paleozoic amniotes, herbivores had dramatically more rapid rates of tooth replacement, with extremes in herbivorous taxa like *Diplodocus* (35 days) at one end of the spectrum, while large carnivores like *Tyrannosaurus* (777 days) were at the other end of the spectrum[28]. What appears to differentiate the Mesozoic herbivorous and carnivorous dinosaurs from Paleozoic amniotes appears to be at least in part related to feeding behaviour. The herbivorous dinosaurs with the fastest replacement rates evolved relatively small individual teeth organized into dental batteries and a strategy of rapid replacement rates associated with extensive dental wear and consumption of highly corrosive food matter, and extensive oral processing[34]. At the other end of the spectrum, the large apex predatory dinosaurs evolved massive dentitions, the largest teeth known in terrestrial vertebrate evolution, and show adaptations against tooth breakage[34,35].

The opposite trend was observed for the early Permian synapsids, with the herbivorous taxa having relatively slow rates of replacement and the carnivorous taxa having rapid rates of tooth replacement. The unexpectedly rapid replacement rates of *Mesenosaurus* may reflect a strategy for ensuring the continued maintenance of efficient dentition in a very gracile predator. The replacement rates for two extant taxa also included in this study for comparison may provide some insights here. The carnivorous Bengal monitor (*Varanus bengalensis*) is similar in size to *Mesenosaurus* with similarly recurved teeth but without serrated carinae[36]. Interestingly, the replacement rate for *V. bengalensis* was similar to that of the early Permian reptiles, as was the replacement rate in other extant reptiles like *Iguana iguana*[37], *Alligator*, and *Caiman*[19].

*Mesenosaurus* teeth morphologically resemble those of the extant Komodo dragon (*Varanus komodoensis*) in being slender, labiolingually compressed, with narrow crowns, strong curvature, and serrated carinae (ziphodonty). This similarity is probably indicative of similar feeding strategies and dental function, in particular defleshing of large prey. In this context, we found that a tooth of *V. komodoensis* is only functional for approximately 92 days (from the initial ankylosis with the jawbone to shedding), with rapid replacement rates as each tooth position has multiple replacement teeth in place. The teeth are ideally suited for the observed defleshing feeding behaviour of this predator while avoiding extensive tooth-bone contact[38], and we suggest that *Mesenosaurus* likely had similar feeding behaviour as adults.

Our study shows that patterns of tooth development and replacement represent two different but related phenomena, the rate at which a new tooth is produced by the dental lamina and the longevity of the functional tooth prior to shedding. Although continuous tooth replacement appears to be universal among early terrestrial vertebrates, including amniotes, our study shows remarkable variation in rates of tooth replacement and longevity, even during the initial stages of amniote diversification, the first terrestrial vertebrates to adapt to surprisingly varied feeding strategies.

In particular, our results show a strong correlation between the feeding behaviours of various Paleozoic synapsids (herbivores and various types of predators) and their tooth replacement rates, where we observe three distinct categories: long-lived teeth with slower replacement rates, as seen in herbivores; long-lived teeth with relatively more rapid replacement rates, as seen in the apex predator *Dimetrodon*; and short-lived teeth with rapid replacement rates, as in the varanopid predator *Mesenosaurus*.

Early reptiles and stem amniotes may represent the primitive condition for amniote patterns of tooth replacement, not significantly different from most extant reptiles. Departures from this presumed primitive amniote condition appear to be related to the diversification of feeding behaviours exhibited by herbivorous and carnivorous amniotes, with herbivorous synapsids greatly reducing the rates of tooth replacement and increasing significantly the longevity of their functional teeth. This is in keeping with their diet and the need to process resistant food materials, irrespective of the presence or absence of tooth-on-tooth contact. In contrast, the predators appear to undertake two different strategies, with the large apex predators (*Dimetrodon*) maintaining the presumed primitive rate of tooth replacement, the timing of the formation of new teeth, but greatly increasing the longevity of the functional teeth. This is likely related to the need for increasing the thickness and massiveness of their teeth for capturing large prey. In contrast, the smaller gracile *Mesenosaurus* took a dramatically different approach, having unusually rapid rates of tooth replacement, and dramatically reduced longevity of their functional teeth. This is confirmed by the common occurrence of shed teeth of this predator in the cave deposits at Richards Spur.

## Methods

This research complies with all relevant ethical regulations. The materials used in the present study consist of isolated jaw elements of the following early Permian synapsid taxa: *Mesenosaurus efremovi* and *Oromycter* from the early Permian Richards Spur locality (Dolese Brothers Limestone Quarry), Oklahoma, USA; *Watongia meieri* from BC7 (Olson 1965), Blaine County, Oklahoma; *Dimetrodon* cf. *D.limbatus* from Briar Creek Bonebed, Texas; Haptodontine from the Lansing Group, Kansas, Anderson County, USA; *Haptodus* sp. from the Garnett Quarry, located in Anderson County, Kansas; *Edaphosaurus* sp. from Hog Creek, Arroyo Formation, Clear Fork Group, Texas; and *Ennatosaurus tecton* from the middle Permian Moroznitsa locality, Mezen, Russia. In addition, isolated jaw elements of the eureptiles *Opisthodontosaurus* and *Captorhinus*, the parareptiles *Delorhynchus* and *Colobomycter*, and the seymouriamorph *Seymouria* were used from the early Permian Richards Spur locality. The paleogeographic distribution of these taxa is shown in Fig. 6. The jaw elements were photographed labially and lingually using Leica DVM6 digital microscope and LAS X software. Morphological measurements for tooth height (TH) were determined using the corresponding measurement tool in the ImageJ software version 2.0.0.

### Histology

Three types of thin sections of marginal dentition were constructed at the University of Toronto Mississauga: a single longitudinal anteroposterior (AP) section of USNM PAL 706602, ROMVP 85526, and ROMVP 85527; four longitudinal labiolingual (LL) sections on ROMVP 85511; three LL sections of ROMVP 85457; two LL sections of ROMVP 85455, UMCP 143278, ROMVP 85516, ROMVP 855214, and ROM37036; one LL section of ROMVP 85453, ROMVP 85502, ROMVP 85503, ROMVP 85504, ROMVP 85505, ROMVP 85506, ROMVP 85507, ROMVP 85508, ROMVP 85509, ROMVP 85510, *Hap*-UTM-001, USNM PAL 706602, PIN 4543, ROMVP 85512, ROMVP 85513, ROMVP 85521, ROMVP 85521, ROMVP 85515; and ROM R271; two transverse (TR) sections on ROMVP 85443 and two of ROMVP 85445. The histological thin sections were produced through partial destructive sampling with permission from the relevant institutions.

We were able to cut the specimens exactly transverse to the jaw in order to avoid the appearance of artificially greater widths of increment lines that has been previously shown to occur[39]. The specimens were first individually embedded in Castolite AC polyester resin, placed inside a vacuum and left to cure for a minimum of 24 hours before cutting. After the resin hardened, the specimens were cut using the Metcut-5 low-speed wafer blade saw (225 rpm), and the cut surfaces were polished using 1000-grit silicon carbide paper with water as a lubricant. The polished surfaces of the specimens were then mounted on frosted plexiglass slides using super glue. Once the glue dried, the mounted specimens were cut again using the Metcut-5 resulting in the mounted specimens having a thickness of approximately 1 mm.

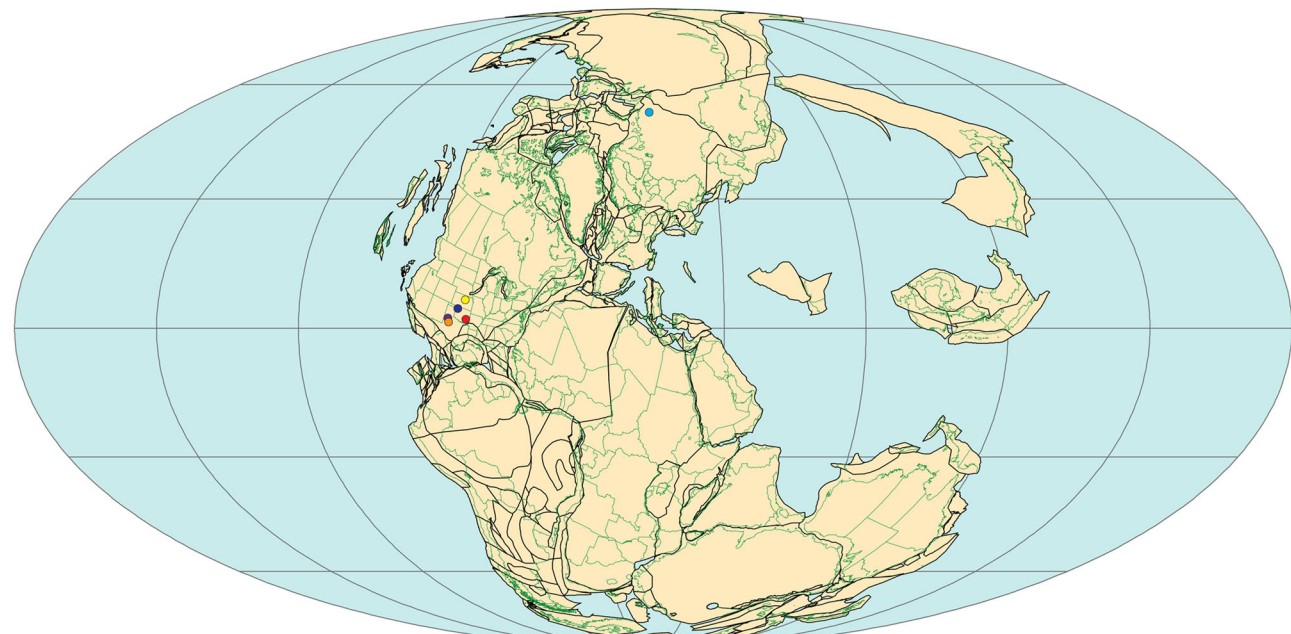

**Fig. 6 | Paleogeographic distribution of the fossil taxa in Late Paleozoic Pangea.** The amniotes *Mesenosaurus, Oromycter, Opisthodontosaurus, Captorhinus, Delorhynchus, Colobomycter,* and the stem amniote *Seymouria* were collected at the Dolese quarry, Richards Spur, Oklahoma, USA (red dot); the varanopid *Watongia* was found in Blaine County, Oklahoma (blue dot); the sphenacodontid *Dimetrodon* was collected at Briar Creek Bonebed, while *Edaphosaurus* was found at Hog Creek, Arroyo Formation, Clear Fork Group, Texas (orange and brown dots); the haptodontine and *Haptodus* were found at the Garnett Quarry, Anderson County, Kansas (white dot); *Ennatosaurus* is from the Moroznitsa locality, Mezen, Russia (black dot). Map illustrated by The PLATES Project, Institute for Geophysics, University of Texas at Austin.

The mounted specimens were then ground to approximately 100 μm using a Metcut-10 Geo (MetLab) machine grinding with a 30 mm grinding cup. Each thin section was further ground by hand using 1000-, 1500-, and 2000-grit silicon carbide paper with water as a lubricant and hand-polished using 1 mm grit aluminium oxide powder on a felt cloth. The thin sectioned specimens were photographed using a Nikon DS-Fi1 camera mounted onto a Nikon AZ-100 microscope with a magnification between 5 × 1 and 5 × 7.

### Replacement rate

Once photographed, the replacement rate of marginal dentition was assessed using thin sections of tooth families from nine maxillae (ROMVP 85443, ROMVP 85445, ROMVP 85455, ROMVP 85457, ROMVP 85491, ROMVP 85502, ROMVP 85503, ROMVP 85504, and ROMVP 85507) and four dentaries (ROMVP 85505, ROMVP 85506, ROMVP 85508, and ROMVP85509) for *Mesenosaurus efremovi*; one maxilla for *Dimetrodon* (ROMVP 85510), *Oromycter* (ROMVP 85516), *Edaphosaurus* (USNM PAL 706602), *Delorhynchus* (ROMVP 85513), *Opisthodontosaurus* (ROMVP 85511), and *Varanus bengalensis* (ROM R271). Additionally, one dentary for *Seymouria* (ROMVP 85515) and two for *Delorhynchus* (ROMVP 85512 and ROMVP 85514); one premaxilla for *Colobomycter* (ROMVP 85521) and *Captorhinus* (ROMVP 85525) were sectioned. A tooth family consists of a functional tooth and its replacement tooth that develops lingually with respect to the functional tooth, which both arise from the same germinal material[18,40].

The thin sections used to analyze replacement rates were constructed and photographed using the method outlined above. Incremental lines of von Ebner were identified on the thin sections based on previously noted characteristics such as single increment widths ranging from 1 and 30 μm, alternating opaque zones, and line trajectories perpendicular to dentinal tubules and aligned at an angle to the depositional plane created by the enamel-dentine junction[18,19]. The average widths of the incremental lines in each section were determined using measurements obtained by image analysis on ImageJ,

version 2.0.0. Incremental lines were not always visible throughout an entire section, possibly as a consequence of fossilization or non-uniform sanding; therefore, in each section, the longest sequence of von Ebner lines was measured perpendicular to the deposition plane. The distance of the sequence was divided by the respective number of increments that spanned it to determine the mean line width. Determining the mean increment width was also necessary due to slight variation between individual increment widths within a specimen. For TR cross-sections, the total number of increments was determined from the pulp cavity to the tooth exterior in a trajectory perpendicular to the plane of deposition (Supplementary Fig. 1b). For longitudinal LL sections, counts were made from the pulp cavity to the tooth exterior following an incremental line, in a trajectory perpendicular to the plane of deposition, towards the exterior end of the tooth. This was repeated until the total number of increments was recorded (Supplementary Fig. 1d). Once the total incremental lines were determined for a tooth family, the rate of tooth replacement at each position was calculated as the difference between the total number of days a functional tooth has been growing for and the total number of days that the replacement tooth has been growing[18,19]. The total number of increments in a functional tooth, which is theoretically is equivalent to the total number of days a tooth has been growing, based on previously mentioned studies, was subtracted by the total number of increments in its immediate replacement tooth to estimate the replacement rate, in days, for each tooth family.

To estimate the age of the lost replacement tooth, the height of the resorption pit (RP) on the functional tooth was measured, and the approximate height of the replacement tooth was calculated to be two-thirds of the RP height (Supplementary Fig. 1c). The estimated height was overlaid on the exterior edge of the functional tooth to count the incremental lines of von Ebner starting from the apex to the last line within the area encompassed by the estimated height. Once the approximate age of the replacement tooth was identified, it was subtracted from the age of the functional tooth to calculate the

replacement rate. This method was used for *Opisthodontosaurus* (ROMVP 85511), *Ennatosaurus* (PIN 4542), *Edaphosaurus* (USNM PAL 706602), and *Seymouria* (ROMVP85515). The method for the approximation of replacement rate was first utilized on specimens with intact replacement teeth of known age. This allowed for comparing the approximation to actual tooth age to confirm the method's accuracy. The method approximated the replacement rate with slight variation; thus, an expected margin of error is associated with the results.

The body mass (kg) of the taxa were estimated from the specimens histologically sampled, with reference to Romer and Price[41] and comparisons with mass of equivalent-sized extant squamates.

### Reporting summary

Further information on research design is available in the Nature Research Reporting Summary linked to this article.

### Data availability

The histological thin section taxa data used in this study are available in the Morphobank database with the link: http://morphobank.org/permalink/?P4321. Additionally, source data is provided with this paper. For access to the original materials, please contact Dr. David Evans at the Royal Ontario Museum; Dr. Hans Sues at the National Museum of Natural History Smithsonian Institution; Dr. Andrey Sennikov at the Paleontological Institute, Russian Academy of Science. Source data are provided with this paper.

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

## Acknowledgements
We thank Bill May for continued support and donation of specimens used in this study, Hans Sues and Andrey Sennikov for providing access to materials, Luke Mahler for discussion and assistance with R Studio. Jilin University and NSERC Discovery Grant provided support for R.R.R., NSERC CGS-M for T.M. and S.M.

## Author contributions
R.R.R. conceptualized the study. T.M. and S.M. completed the histological analysis, analyzed the data, and prepared figures. T.M. wrote the initial draft of the manuscript. D.S. provided the skull reconstruction, fossil preparation, and photography of specimens. All authors contributed and approved the final draft of the manuscript.

## Competing interests
The authors declare no competing interests.
