## [Peer Review File · Nature Communications]

Permian hypercarnivore suggests dental complexity among early amniotesReviewers' Comments:

Reviewer #1:

Remarks to the Author:

I enjoyed reading this well-designed, data-rich study. The authors were clear in what they did and put it into an interesting phylogenetic context. The results are intriguing, important for a wide audience, and will spur further research. I have made comments and edits using track changes in MS word (attached). I have a few more involved suggestions, outlined below:

- 1) Body size or mass is only casually mentioned a few times throughout the manuscript, but it really should be quantified and results should be contextualized with it. Where this is most needed is Figure 5b – though I think that incremental line width should be plotted and that graph included (perhaps as supplemental information), the more appropriate plot would be mapping incremental line width divided by body mass. D'Emic and Finch (2022, *Biology Letters*) provided some coarse body mass estimates using a variety of methods for varanids that could be applied to taxa here with a similar bauplan.
- 2) Also, Finch and D'Emic (2022; *Biology Letters*) provided a phylomorphospace which was incompletely filled; your new data seem to fill in the missing part of that phylomorphospace, so you could mention that here.
- 3) In the spirit of reproducibility and open science, it would be good to upload high-resolution images (jpgs or tiffs) of full histological images of each specimen to a repository like Morphobank.
- 4) Transverse sections can produce obliquely cut incremental lines, making them appear artificially thicker than they are (Kosch and Zanno, 2020: *PeerJ*). This should be mentioned and discussed for your transversely oriented sections.
- 5) Typo in genus name in supplemental info: "Extended Data Figure 5. Incremental lines of Ophisthodontosaurus."

Reviewer #2:

Remarks to the Author:

This manuscript is a very interesting one, incorporating established methodology and previously verified concepts, using them in a novel approach to a much older fauna, taking advantage of an unusual abundance of newly discovered fossil material. It may not be particularly crucial for the evolution of jaws and teeth in vertebrates, but it is an important addition to our understanding of the food webs among the first terrestrial vertebrate communities. I suggest it to be published with minor revisions.

The text is well written, aims are clear, concise and the ideas well expressed. The definitions are precise, the methodology is clear and comprehensible. The conclusions are supported by the data. I expect the paper to be of interest not only for vertebrate palaeontologists, but also for experts in various areas of biological research.

Below, please find some minor comments and suggestions that I believe could help the authors improve the clarity of some parts of the manuscript:

Line 18: "among them" Among whom? Do they belong to reptilian or synapsid amniotes? I know that, but the reader may not. You should mention the systematic position of Mesenosaurus at least once in the manuscript;

Line 31: "likely representing the primitive condition" On what grounds do you assume that? Please specify (here or elsewhere in the text);

Line 42-44: "extensive modifications" vs. "extensively modified" Too many "extensive modifications" here, clarity is lost, please rephrase;

Line 57: "Among them" Again, among whom?;

Line 109: Missing dot after parenthesis;

Line 115-120: Not all the mentioned genera are listed here, why? How was the longevity/replacement rate of e.g. Haptodus or Watongia calculated?;

Line 250: "serrated" I wish I could see that in detail on the photographs;

Line 306: replace "for" with "of";

Line 422: can you add "RP" to the figure?;

Line 455: replace "Sci. Adv" with "Science";

Extended data figures: I would welcome a locality map (either on a paleo or a recent map base) of specimens histologically sectioned for the manuscript. The authors keep mentioning vertebrate communities throughout the text, but in what sense? Purely coeval or also geographical?

Valéria Vaškaninová

Response to reviewer's comments

Reviewer #1

We agreed with all of the edits made directly to the word document and the changes and additions recommended in the Remarks to the Author. We have made the corresponding changes to the revised manuscript, as listed below.

Commented [A1]: This is somewhat confusingly worded, because it implies that there were non-gracile, non-hypercarniverous varanopids before the Carboniferous.

- Agree, took out “gracile”

Commented [A2]: I’d suggest rephrasing this clause, if not the whole sentence. As written, it sounds like this was work done in the past that you are introducing to set up the present study, when in fact it is the results of the present study.

- Agree, rephrased the sentence

Commented [A3]: This is confusing, because lots of vertebrates have serrated teeth (theropods, some felids...).

- We rephrased the sentence. The sentence was meant to refer only about living species.

Commented [A4]: This transition makes this sentence seem like an afterthought or corollary of your study, when it is part of the bulk of its conclusion.

- Agree, rephrased the sentence and took out the word transition.

Commented [A5]: Perhaps change this to “whereas”, because “while” implies simultaneous processes. The evolutionary timing of these events doesn’t seem to be pinned down enough to say it occurred simultaneously.

- Agree, changed to “whereas”

Commented [A6]: In the paper, you map traits onto a phylogeny...doesn’t that give you a sense of the primitive condition?

- Yes it does. We rephrased the sentence to make it clearer.

Commented [A7]: Ossification generally refers to bone, so perhaps rephrase this to “mineralization” or “deposition”

- Agree, changed word to “mineralization”

Commented [A8]: Citation needed at the end of this sentence of after “Erickson”

- Added citation

Commented [A9]: Rather than ROM, ROMVP or Dim-ROMVP, Mes-ROMVP, etc. are listed below

- Fixed this.

Commented [A10]: Not necessarily;

- Agree, we removed this section.

Commented [A11]: Deleted because it was repetitive

- Agree

Commented [A12]: This is methods, not results.

- Agree, we removed this section.

Commented [A13]: It is confusing that some of the specimen numbers begin with the first three letters of the genus name, and some don't. None of the latter are in the "institutional abbreviations" listed above.

- Agree. We removed the first three letters of the genus name to keep it consistent.
- Added the institutional abbreviations that were missing.

Commented [A14]: This reads like methods, interspersed between two sentences reporting results.

- Agree. We rephrased the sentence to remove the methods and only discuss the results.

Commented [A15]: How was ontogenetic stage determined? By size alone? Size is a poor predictor of ontogenetic status in many extant reptile groups (see Griffin et al.'s 2021 review in Biological Reviews)

- The statement was for one specimen (no variation of replacement rate in tooth position, size, or ontogenetic age of tooth). We rephrased the sentence to highlight this point more clearly.

Commented [A16]: Again, this paragraph is methods, not results.

- Agree, we removed this section.

Commented [A17]: I deleted "surprisingly", because that sort of interpretive context better fits in the discussion, not results

- Agree

Commented [A18]: Technically, 152 days is the minimum replacement rate if no replacement teeth are present.

- Agree, changed the sentence to reflect this point

Commented [A19]: This phrasing is confusing, making it sound like Mesenosaurus is similarly "highly endangered".

- Agree, moved "highly endangered" to be after the Komodo dragon.

Commented [A20]: Why all of the sudden are fractions of a line reported?

- Changed to whole numbers

Commented [A21]: I can see $227-135 = 92$, but tooth replacement rates vary, so why not also report $227-106 = 121$ days? And average 92 and 121 to give 107 days as the average rate for *V. komodoensis*?

- Agree. Rephrased the sentence to include the average of 107 days for the Komodo.

Commented [A22]: Why does this specimen number begin with Dim?

- It doesn't begin with Dim. We changed all of the specimen numbers and removed the first three letters of the genus.

Commented [A23]: I would recommend using one term and sticking with it. You use lines of von Ebner, growth lines of von Ebner, incremental growth lines of von Ebner, and here von Ebner incremental markings.

- Agree

Commented [A24]: This contradicts what was stated above, where you say you didn't observe a difference among specimens of different ontogenetic stage.

- For the statement above [Comment A15] we were talking about variation within a single tooth row for an individual specimen, comparing newly replaced teeth with the old teeth.
- For this statement [A24], we are talking about multiple specimens, since we tried to eliminate the confounding factor of age by using maxillae and dentaries of similar size. However, we accept that fact that size is not a good indicator of assuming ontogenetic stage.

Commented [A25]: This doesn't quite make sense. Tooth replacement is the former, whereas tooth longevity is the latter. These are different.

- Agree. We rephrased the sentence to clear it out.

Commented [A27]: Incremental lines of von Ebner are not parallel to the EDG, they are aligned at an angle to it.

- Agree, changed the wording of the sentence.

1) Body size or mass is only casually mentioned a few times throughout the manuscript, but it really should be quantified and results should be contextualized with it. Where this is most needed is Figure 5b – though I think that incremental line width should be plotted and that graph included (perhaps as supplemental information), the more appropriate plot would be mapping incremental line width divided by body mass. D'Emic and Finch (2022, Biology Letters) provided some coarse body mass estimates using a variety of methods for varanids that could be applied to taxa here with a similar bauplan.

- Agree. We added the graph of replacement rate compared to body mass of taxa to supplementary material since it does not show a significant correlation.

2) Also, Finch and D'Emic (2022; Biology Letters) provided a phylomorphospace which was incompletely filled; your new data seem to fill in the missing part of that phylomorphospace, so you could mention that here.

- Agree. Added this information to the Discussion.

3) In the spirit of reproducibility and open science, it would be good to upload high-resolution images (jpgs or tiffs) of full histological images of each specimen to a repository like Morphobank.

- Agree. We will be uploaded the images to Morphobank.

- 4) Transverse sections can produce obliquely cut incremental lines, making them appear artificially thicker than they are (Kosch and Zanno, 2020: PeerJ). This should be mentioned and discussed for your transversely oriented sections.
 - Agree. We added this information to the Methods section.
- 5) Typo in genus name in supplemental info: “Extended Data Figure 5. Incremental lines of Ophisthodontosaurus.”
 - Fixed the typo for genus name → *Opisthodontosaurus*

Reviewer #2

We agreed with all of the edits recommended in the Remarks to the Author. We have made the corresponding changes to the revised manuscript, as listed below. However, we do not feel that the suggestion of adding a locality map in supplemental materials is practical or useful, as indicated below.

Line 18: “among them” Among whom? Do they belong to reptilian or synapsid amniotes? I know that, but the reader may not. You should mention the systematic position of *Mesenosaurus* at least once in the manuscript;

- Agree with this comment. Added “among the early synapsids”

Line 31: “likely representing the primitive condition” On what grounds do you assume that? Please specify (here or elsewhere in the text);

- We rephrased this sentence to make it more clear. The primitive condition was assumed by determining the general condition found in the stem amniote, the two parareptiles, and compared to extant varanid lizards.

Line 42-44: “extensive modifications” vs. “extensively modified” Too many “extensive modifications” here, clarity is lost, please rephrase;

- Agree with this comment. Rephrased the sentence to have “changes” rather than “extensive modifications”

Line 57: “Among them” Again, among whom?;

- Agree with this comment. Added “among the early synapsids”

Line 109: Missing dot after parenthesis;

- Fixed this.

Line 115-120: Not all the mentioned genera are listed here, why? How was the longevity/replacement rate of e.g. *Haptodus* or *Watongia* calculated?;

- Removed this sentence since the 1st reviewer mentioned that it was describing the methods (in the results section).

Line 250: “serrated” I wish I could see that in detail on the photographs;

- We included a supplementary photograph with the serrations. Also, we are currently working on a paper showing the unique serrations for *Mesenosaurus*.

Line 306: replace “for” with “of”;

- Fixed

Line 422: can you add “RP” to the figure?;

- Added “RP” to the supplementary figure 1

Line 455: replace “Sci. Adv” with “Science”;

- Fixed

Extended data figures: I would welcome a locality map (either on a paleo or a recent map base) of specimens histologically sectioned for the manuscript. The authors keep mentioning vertebrate communities throughout the text, but in what sense? Purely coeval or also geographical?

- Most of the taxa are both coeval (Early Permian) except for the Russian *Ennatosaurus* (Middle Permian) and the majority of the specimens were collected at Richards Spur, Oklahoma, as already explained in the methods section. We feel that a locality map at the global scale would not be helpful beyond the information already provided in the methods section (One locality in Oklahoma, one in Kansas, two localities in Texas, and one locality in Russia).

Reviewers' Comments:

Reviewer #1:

Remarks to the Author:

The manuscript is now in great shape and ready to be published. This is an interesting and important contribution.

Reviewer #2:

Remarks to the Author:

Thank you for the revised manuscript. I am now content with it, no need for further edits.

However, I may have missed it, but I did not find "a supplementary photograph with the serrations" (as the authors write in their response) within the text figures nor the supplementary files. I am still convinced that a detailed photograph of the serrated tooth margin should be included, since it is so often mentioned in the study and represents an important part of its claims.

We included a supplementary photograph with the serrations as Supplementary Figure 8.

Also, we are currently working on a paper showing the unique serrations for Mesenosaurus.